# CUSTOMIZING GLOBAL MODEL FOR ARBITRARY TARGET DISTRIBUTIONS IN FEDERATED LEARNING

## ABSTRACT

Federated learning (FL) is a privacy-preserving approach to train a global model on decentralized data. Most existing FL algorithms optimize the global model by minimizing the average loss among clients, aiming to perform well on commonly assumed uniform target data distribution. In practice, though, the need often arises for a tailored model to excel on its specific unlabeled target dataset with arbitrary distribution. The misalignment of the assumed and actual target distribution violates the plausible uniform assumption and thus undermines the effectiveness of vanilla FL methods. To fill this gap, we propose FedSSA, a self-supervised aggregation method capable of training a specific global model for specific target data. FedSSA leverages the target dataset on the server side to dynamically learn aggregation weights for local models in a self-supervised manner. These aggregation weights are iteratively adjusted to promote transformation-invariant. With extensive qualitative and quantitative experiments, we demonstrate that FedSSA consistently outperforms 12 classical baselines across multiple datasets, heterogeneity scenarios and different target distributions. Furthermore, we showcase the plug-and-play property of FedSSA by combining it with various FL methods.

## 1 INTRODUCTION

As a privacy-preserving training paradigm, federated learning (FL) (McMahan et al., 2017) aims to collaboratively train a global model across multiple clients without exchanging private data. It has been widely applied to various tasks, including computer vision (Hsu et al., 2020; Jiménez-Sánchez et al., 2023; Li et al., 2019a; Chen et al., 2023; Liu et al., 2020), natural language processing (Yang et al., 2018; Wang et al., 2020a) and Internet of Things (IoT) (Campos et al., 2022; Friha et al., 2022). There are many bottleneck issues that hinder the further development of FL, like data heterogeneity Gao et al. (2022a); Tan et al. (2022); Zhao et al. (2018); Li et al. (2019b), communication and computation efficiency (Li et al., 2020a; Almanifi et al., 2023).

When addressing these issues, most existing FL algorithms tend to make assumptions about the target distribution. Some works assume it to be uniform (McMahan et al., 2017; Chen & Chao, 2022) or exactly the same as the overall training distribution (Li et al., 2020b; Karimireddy et al., 2020; Li et al., 2021) (defined as the union of client training distributions). Another line of work, such as AFL (Mohri et al., 2019), optimizes any target distribution consisting of a mixture of client distributions, fostering fairness and addressing distribution intricacies. Unfortunately, the target distribution could be arbitrarily different from the overall training distribution in practice, rendering conventional FL methods ineffective.

This paper delves into the targeted training scenarios within the FL paradigm dubbed the *Pragmatic Federated Learning*. Pragmatic Federated Learning is a general and pragmatic approach for model customization within the FL paradigm, enabling efficient and effective adaptation to diverse target datasets while preserving privacy. Consider the setting in model markets (Vartak et al., 2016; Zhang et al., 2022a), individual buyers possess unlabeled data and require well-performing models explicitly tailored to their unique target data. However, practical limitations, such as inadequate task-specific training data or limited computational resources, make training entirely bespoke models for target data infeasible, requiring the utilization of federated learning. These practical scenarios highlight the immense potential of Pragmatic Federated Learning as a solution for customizing well-performing models to specific target data.

The crux of the proposed Pragmatic Federated Learning, lies in assigning appropriate weights to local models to align the training distributions with a specific target distribution. Obviously, traditional target-agnostic weight allocation caters to overall training distribution and thus, fails to deliver well-performing customized models for arbitrary target distribution. This is further complicated by two factors: (1) the limited insight into distributions of clients due to the privacy restrictions in FL (Luo et al., 2021; Zhang et al., 2022c; Yuan et al., 2022) and unlabeled target data, and (2) the inability to assess local models on unlabeled target data. Thus, these two facts make it a critical challenge to customize an aggregated model for this arbitrary target distribution, which motivates the exploration of more flexible and pragmatic FL paradigms that adapt to the diverse target distribution.

To tackle this challenge, our core idea is to use the self-supervised technique to guide the model ensembling. As self-supervised learning leverages intrinsic information underlying the target data itself to create surrogate tasks, we can capture the representativeness of local models on target data and assign the aggregation weights. In essence, by adjusting aggregation weights, self-supervised learning optimizes the ensemble learning of local models and enables efficient adaptation to the target distribution.

Following this spirit, we propose a novel pragmatic FL method with **S**elf-**S**upervised **A**ggregation (**FedSSA**), to enhance the flexibility of FL towards different target distributions. In FedSSA, the server fixes local model parameters, aggregates the predictions of local models and learns aggregation weights by promoting transformation-invariant (Mumuni & Mumuni, 2021). Specifically, given two different augmented views of a data sample (Shorten & Khoshgoftaar, 2019), the aggregation weights are tuned by maximizing the cosine similarity between the two overall predictions, which are obtained by aggregated predictions of local models. Focusing exclusively on transformation invariance may lead to weights concentrated on a single local model. To counteract this effect, we design to maximize prediction confidence to prioritize more reliable clients, and we maximize the weight entropy to encourage multi-agent collaboration. Without prior insight into data distributions, FedSSA ensures ensemble of local models aligns effectively with the target distribution.

The distinct advantages of FedSSA include (1) Customized Global Models: FedSSA excels at tailoring global models to specific target data distributions. (2) Efficient Weight Learning: Under the guidance of unlabeled data, FedSSA efficiently learns aggregation weights that promote transformation-invariant, confident predictions and multi-client collaboration.

We conduct extensive experiments to show that FedSSA consistently achieves the best by comparing with 12 classical baselines on four classical datasets, under two common types of training data heterogeneity (McMahan et al., 2017; Wang et al., 2020b), and three representative target data distributions. Besides, we demonstrate the modularity of FedSSA by showing that FedSSA can be easily applied with various existing FL methods to further enhance their performance. Furthermore, in cases where the unlabeled dataset is unavailable during training, FedSSA still shows advantages for per-data-sample prediction during inference.

In summary, our main contributions are as follows:

1. We introduce the concept of Pragmatic Federated Learning, a versatile approach under the FL paradigm. Pragmatic Federated Learning offers a practical solution for efficiently adapting global models to diverse target datasets while preserving privacy;

2. We propose FedSSA, a novel self-supervised learning scheme at the server side in FL, which learns aggregation weights by promoting transformation-invariant and confident model prediction while pursuing multi-client collaboration;

3. We conduct extensive experiments to show better performance and the modularity property by comparing with 12 classical baselines.

## 2 RELATED WORK

**Federated learning.** Federated learning (FL) enables multiple clients to collaboratively train a global model without sharing private data (McMahan et al., 2017) and has been widely applied to diverse fields, including image classification (Hsu et al., 2020), medical analysis (Liu et al., 2021a) and object detection (Liu et al., 2020).

*Data heterogeneity.* The training distributions of clients could distinctly vary since data are collected under various circumstances and preferences (Kairouz et al., 2021; McMahan et al., 2017). Addressing this issue, many FL variants have been proposed from two aspects: global model adjustment and local model correction. (1) FL on global model adjustment. To enhance consistency among local models at the client side, such as applying $\ell_2$-based model regularization between local and global model (Li et al., 2020b; Acar et al., 2020) and introducing a correction term on the model gradient (Karimireddy et al., 2020; Gao et al., 2022b). (2) Improving performance of the global model at the server side, such as applying momentum-based global model updating (Hsu et al., 2019; Reddi et al., 2020) and knowledge distillation to tune the global model (Lin et al., 2020; Li & Wang, 2019; Chen & Chao, 2020). Our work is orthogonal to them and can be easily combined with them to adaptively learn aggregation weights to further enhance the overall performance.

*Aggregation weights learning.* In the most widely used strategy (McMahan et al., 2017; Zhang et al., 2022b; Li et al., 2021), the aggregation weights are proportional to the size of local data, and cater to the overall training distribution. These vanilla FL methods fail to customize the global model for specific target data. Some works (Nishio & Yonetani, 2019; Guha et al., 2019) determine their weights by the loss-based selection, which requires the validation dataset performance, not applicable to unlabeled data. Moreover, Reyes et al. (2021) propose inverse variance estimates to learn weights, which adds communication and computation costs. AFL (Mohri et al., 2019) introduces a centralized model optimized for any target distributions. It simultaneously modifies the local model training and aggregation weights for the global model. However, AFL relies on labeled data to leverage loss information for adjusting aggregation weights through the projection property.

**Self-supervised learning.** Generally, self-supervised learning aims to train a model to capture general patterns from the massive unlabeled data (Chen et al., 2020; He et al., 2022), where a common solution is to pursue transformation-invariant features given two different augmented views of the same image (Grill et al., 2020; Chen & He, 2021). Recently, self-supervised tasks have been discussed in the context of FL. For example, FedU (Zhuang et al., 2021), FedEMA (Zhuang et al., 2022), FedX (Han et al., 2022) and FedVSSL (Rehman et al., 2022) consider federated self-supervised model pre-training. While all these methods aim at tuning all model parameters via self-supervised learning (Zbontar et al., 2021; He et al., 2020), we apply self-supervised learning only to tune aggregation weights while keeping the model parameters fixed.

As a sub-field, test-time training aims for tuning parameters based on unlabeled samples during inference time (Sun et al., 2020; Liu et al., 2021b; Wang et al., 2021a). SADE (Zhang et al., 2022d) learns weights for a triple-branch network for the long-tailed scenario, where each branch is trained under a known pre-defined distribution and evaluates under several known pre-defined distributions; while we consider a more challenging task where the number of branches is not fixed, both training and evaluating distributions are unknown in the setting of Pragmatic Federated Learning.

## 3 PROBLEM FORMULATION

Suppose in an FL system, there are $K$ clients in total and each client $k$ holds a private dataset $\mathcal{D}_k = \{(\boldsymbol{x}_i, y_i)\}_{i=1}^{N_k}$, where $\boldsymbol{x}_i$ and $y_i$ are the raw input and label of $i$-th sample respectively, $N_k$ is the number of data samples. Generally, federated learning methods aim to solve the following minimization problem: $\min_{\boldsymbol{\theta}} \sum_{k=1}^{K} N_k F_k(\boldsymbol{\theta})/N$, where $N = \sum_{k=1}^{K} N_k$ and $F_k(\cdot)$ is the task-driven loss function by evaluating the model on dataset of client $k$. Thus, they often apply the following dataset-size-based model aggregation to obtain the final global model at the server side: $\boldsymbol{\theta} = \sum_{k=1}^{K} N_k \boldsymbol{\theta}_k/N$, where $\boldsymbol{\theta}_k$ is the locally trained model of client $k$.

However, such a global model is not suitable for cases where the target dataset $\mathcal{D}_T$ is drawn from a distribution distinct from the overall training distribution, which is defined as the union of local training distributions of each local dataset $\mathcal{D}_k$. Thus, to alleviate this problem, the weights for aggregating local models should be adjusted such that the global model will be more customized to perform well on the target dataset. Specifically, an optimal global model is defined as:

$$\boldsymbol{\theta}^* = \arg\max_{\boldsymbol{\theta}} P(\mathcal{D}_T, \boldsymbol{\theta}), s.t. \boldsymbol{\theta} = \sum_{k}^{K} \boldsymbol{\theta}_k w_k, \tag{1}$$

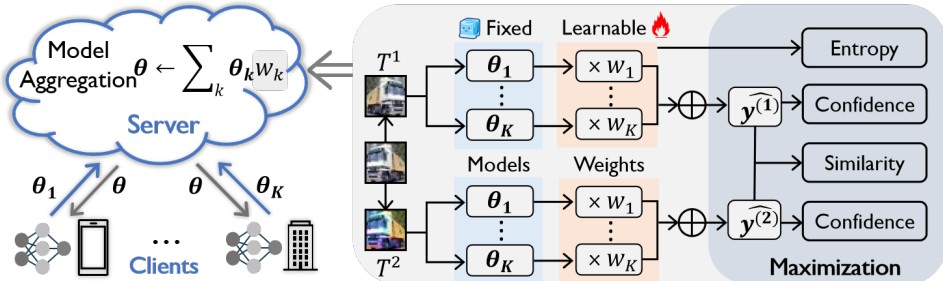

Figure 1: Overview of FedSSA. (1) Each client trains a local model and uploads it to the server. (2) The aggregation weights are learned at the server, where the self-supervised learning process is guided by maximizing the similarity between two final predictions, prediction confidence and weight entropy. (3) The server aggregates local models according to the learned aggregation weights.

where the $P(\cdot)$ indicates some certain task-specific performance evaluation metrics such as the classification accuracy, both local models $\{\boldsymbol{\theta}_k\}$ and $\{w_k\}$ are variables. Unlike most FL algorithms that keep $\{w_k\}$ as fixed by setting $w_k = N_k/N$ or $w_k = 1/N$, we focus on optimizing $\{w_k\}$ for the specific target dataset.

Unfortunately, the target dataset is often unlabeled (denoted as $\mathcal{U}$), and thus the optimization problem in equation 1 can not be directly solved as the performance $P(\cdot)$ cannot be accurately measured when labels are unavailable. Facing this, the aggregation weights should be learned in a self-supervised manner and some self-supervised loss functions should be specifically designed. The original problem equation 1 is then converted to the following objective:

$$\{w_k\} = \underset{\{w_k\}}{\arg\min} \, \mathcal{L}_{SSA}(\{\boldsymbol{\theta_k}\}, \{w_k\}; \mathcal{U}), s.t. \boldsymbol{\theta} = \sum_k^K \boldsymbol{\theta}_k w_k, \qquad (2)$$

expecting the aggregated $\boldsymbol{\theta}$ can better match with the performance of $\boldsymbol{\theta}^*$ obtained in equation 1.

This poses a troublesome challenge due to key issues: (1) no prior knowledge of the local training and target distributions, and (2) unavailability for performance evaluation with unlabeled target data. Addressing these, we design three essential and complementary loss functions (proxy tasks) for self-supervised learning, which learns aggregation weights from the local models and the unlabeled dataset. See theoretical explanation in Appendix A.

## 4 FL WITH SELF-SUPERVISED AGGREGATION

To achieve the goal of customizing an aggregated model that performs well under the target dataset, we propose federated learning with self-supervised aggregation, which leverages the unlabeled dataset to learn the weights for aggregating models at the server side. In this section, we first introduce the overall FedSSA framework, and then, present the learning process of aggregation weights.

### 4.1 OVERALL FRAMEWORK

Following the standard FL procedure, our proposed FedSSA consists of two key steps: local model training and global model aggregation. Our novelty mainly focuses on the second step, where we aim to learn a set of weights customized for the target dataset in model aggregation.

**Local model training.** In the $t$th communication round, the server broadcasts the global (aggregated) model $\boldsymbol{\theta}^t$ to each available client, which is used to synchronize the local model of each client: $\boldsymbol{\theta}_k^t := \boldsymbol{\theta}^t$. Each participating client employs multiple standard SGD steps to update their local model, denoted as $\boldsymbol{\theta}_k^t$, across their private dataset $\mathcal{D}_k$. The local objective is to minimize the task-driven loss $\mathcal{L}_c$, usually the cross-entropy loss for classification tasks. Each step operates as follows: $\boldsymbol{\theta}_k^t := \boldsymbol{\theta}_k^t - \eta_c \nabla \mathcal{L}_c(\boldsymbol{\theta}_k^t; \mathcal{D}_k)$, where $\eta_c$ denotes the learning rate for local model training. Note that we can also apply other local optimizations by adding another loss term, such as a $\ell_2$-distance regularization between local model $\boldsymbol{\theta}_k^t$ and global model $\boldsymbol{\theta}^t$ (Li et al., 2020b).

**Global model aggregating.** After local model training in the $t$th communication round, each client $k$ uploads the local model $\boldsymbol{\theta}_k^t$ to the server. The server then aggregates these local models $\{\boldsymbol{\theta}_k^t\}$ to update the global model $\boldsymbol{\theta}^{t+1}$ for the next round. In conventional FL methods, each aggregation weight is proportional to the corresponding client's dataset size; that is, $w_k^t := \frac{|\mathcal{D}_k|}{\sum_i |\mathcal{D}_i|}$, where $|\mathcal{D}_i|$ denotes the sample number of dataset $\mathcal{D}_i$. However, this aggregation does not take the unlabeled dataset $\mathcal{U}$ at the server side into account. To customize the global model to the unlabeled dataset, we adjust the aggregation weights $\{w_k^t\}$ through self-supervised learning; see details in the following subsection. Then, the global model is obtained by aggregating local models:

$$\boldsymbol{\theta}^{t+1} := \sum_k w_k^t \boldsymbol{\theta}_k^t. \tag{3}$$

## 4.2 Learning Weights via Self-Supervision

To yield a global model that performs well on the target dataset, we propose to learn aggregation weights based on the unlabeled target dataset in a self-supervised manner. Our designed self-supervised learning strategy follows three essential spirits: (1) a stable and powerful model should produce similar predictions for different semantically invariant transformations (such as slight blurring) of the same sample. (2) More confident predictions rather than ambiguous predictions are generally more correct and thus a well-performed model should produce confident predictions. (3) Collaboration among clients is crucial in FL as different local models may have unique insights and beneficial knowledge, so aggregation weight concentrated on a single client should be avoided. Following these spirits, we design three essential and complementary objectives in FedSSA.

**Promoting transformation-invariant prediction.** It is expected that an accurate and stable global model should produce similar predictions for different transformation views of the same sample since the view transformation does not change the semantics. Thus, it is natural to regularize the cosine similarity between the predictions of different views.

Specifically, given a sample $\boldsymbol{x}$ from the unlabeled dataset $\mathcal{U}$, we generate two perturbed views by randomly applying several data transformations (e.g., flipping, Gaussian Blurring, applying color jitters) (Chen et al., 2020; Grill et al., 2020; Zhang et al., 2022d): $\boldsymbol{x}^{(1)} = T^1(\boldsymbol{x}), \boldsymbol{x}^{(2)} = T^2(\boldsymbol{x})$, where $T^1(\cdot)$ and $T^2(\cdot)$ are two transformation functions. The final prediction of each perturbed view is obtained by weighted aggregated predictions of local models given the view as input:

$$\widehat{\boldsymbol{y}}^{(i)} = \sigma(\sum_{k=1}^K w_k \widehat{\boldsymbol{y}}_k^{(i)}), i = 1, 2, \tag{4}$$

where $\sigma(\cdot)$ is the softmax operation, $w_k$ is the aggregation weight for model $k$ and $\widehat{\boldsymbol{y}}_k^{(i)} = h_k(\boldsymbol{\theta}_k; \boldsymbol{x}^{(i)}), i = 1, 2$ is the logits output of local model $\boldsymbol{\theta}_k$ given the $\boldsymbol{x}^{(i)}$ as the input.

As the two perturbed views are generated from the same sample, we expect that their final predictions should be similar. Thus, we aim to maximize their similarity and propose to minimize the following cosine-similarity-based loss $\mathcal{L}_{cos}$ by optimizing the aggregation weights $\{w_k\}$:

$$\mathcal{L}_{cos} = -\cos\left(\hat{\boldsymbol{y}}^{(1)}, \hat{\boldsymbol{y}}^{(2)}\right) = -\frac{\hat{\boldsymbol{y}}^{(1)} \cdot \hat{\boldsymbol{y}}^{(2)}}{\left\|\hat{\boldsymbol{y}}^{(1)}\right\| \left\|\hat{\boldsymbol{y}}^{(2)}\right\|}. \tag{5}$$

Note that only the aggregation weights $\{w_k\}$ are learnable here while the parameters of local models $\{\boldsymbol{\theta}_k\}$ are all kept fixed. Unlike traditional self-supervised learning methods that pursue feature-level alignment (Chen et al., 2020; Grill et al., 2020; Zbontar et al., 2021), we adopt the prediction-level alignment to capture more latent information about data heterogeneity in FL and thus we can learn more appropriate aggregation weights.

**Promoting confident prediction.** Minimizing $\mathcal{L}_{cos}$ promotes transformation-invariant prediction, however, the learning process overlooks the different performances of local models and leads to indistinguishable weights for differently performing models. For instance, a well-performing Model 1 producing confident predictions might be overshadowed by a poorly-performing Model 2 that yields invariant yet irrelevant predictions.

To address this issue, our idea is to encourage confident predictions at the same time. That is, for most samples in a classification task, the probability assigned to one certain category should

be distinctly higher than other categories. Here, we measure the confidence of prediction using variance, that is, the following loss should be minimized:

$$\mathcal{L}_{var} = -\frac{var(\hat{\boldsymbol{y}}^{(1)}) + var(\hat{\boldsymbol{y}}^{(2)})}{2}, \tag{6}$$

where the $var(\hat{\boldsymbol{y}}^{(i)})$ measures the variance among the elements of $\hat{\boldsymbol{y}}^{(i)}$. Minimizing this loss with the $\mathcal{L}_{cos}$ allows the aggregation weights to produce the final prediction with higher confidence while preserving the transformation-invariant property.

**Promoting multi-client collaboration.** As each client in an FL system might have limited data samples to achieve satisfactory performance, collaboration could bring beneficial shareable knowledge (McMahan et al., 2017). For example, if two clients both have similar training distributions with target distribution, a reasonable strategy is to assign them relatively equal aggregation weights. Thus, to pursue multi-client collaboration among clients and avoid aggregation weight concentration during the weights learning process, we propose to minimize the following weight-entropy loss:

$$\mathcal{L}_{we} = \sum_{k=1}^{K} w_k \log w_k. \tag{7}$$

Apparently, when the aggregation weights of all clients are equal, this loss will reach the minimum. So, together with the previous two objectives, this term can contribute to avoiding weight concentration for some specific clients and promote multi-client collaboration.

**Overall SSA loss.** The total SSA loss is defined as:

$$\mathcal{L}_{SSA} = \mathcal{L}_{cos} + \lambda_{var}\mathcal{L}_{var} + \lambda_{we}\mathcal{L}_{we}, \tag{8}$$

where $\lambda_{var}$ and $\lambda_{we}$ are two hyper-parameters to balance the three loss. Note that even when setting $\lambda_{var} = \lambda_{we} = 0$, FedSSA is sufficient to outperform baselines while the last two terms contribute to further performance improvement. By minimizing this overall loss, FedSSA optimizes the aggregation weights $\{w_k^t\}$, which are then applied to aggregate local models uploaded from the clients as equation 3.

## 5 EXPERIMENTS

We compare FedSSA with 12 baselines including local training, FedAvg (McMahan et al., 2017) and several FL variants. Among these, FedProx (Li et al., 2020b), SCAFFOLD (Karimireddy et al., 2020), FedDyn (Acar et al., 2020), MOON (Li et al., 2021), FedDC (Gao et al., 2022b) and Fed-Decorr (Shi et al., 2022) focus on local model correction; FedAvgM (Hsu et al., 2019), FedDF (Lin et al., 2020) and FedExP (Jhunjhunwala et al., 2023) focus on global model adjustment, while AFL (Mohri et al., 2019) adjusts both aspects. Two common heterogeneous settings and several target data distributions are considered. We show key details and results here and leave other implementation details in Appendix B and additional experiments in Appendix C.

### 5.1 IMPLEMENTATION DETAILS

**Data heterogeneity.** We consider four datasets: Fashion-MNIST (Xiao et al., 2017), CIFAR-10 (Krizhevsky et al., 2009), CINIC-10 (Darlow et al., 2018) and CIFAR-100 (Krizhevsky et al., 2009), which are frequently used in FL literature (Zhang et al., 2022a; Shi et al., 2022; Jhunjhunwala et al., 2023). For data heterogeneity, we consider two heterogeneous (non-IID) settings, namely NIID-1 and NIID-2. Considering a $C$-classification task, NIID-1 is an extremely severe heterogeneous setting, where each client holds data from $C/5$ categories (Li et al., 2022; McMahan et al., 2017). NIID-2 follows Dirichlet distribution $Dir_\beta$ (default $\beta = 0.5$) (Wang et al., 2020b; Yurochkin et al., 2019; Acar et al., 2020; Li et al., 2021), where smaller $\beta$ implies more severe heterogeneity.

**Target distribution.** For both two heterogeneous settings (NIID-1, NIID-2), the union of training category distributions of all clients is uniform. While most existing FL methods only consider uniform target distribution, we consider diverse target distributions: skew, imbalanced, and uniform distribution. The skew distribution aligns with the training distribution of Client 1. The imbalanced distribution follows an exponential distribution (Cui et al., 2019): $n_c = n_1 \rho^{-\frac{c-1}{C-1}}$ is the number

Table 1: Classification accuracy (%) Comparison of Classification Accuracy (%) under NIID-1 on Fashion-MNIST, CIFAR-10, CINIC-10, and CIFAR-100 with three target distributions (TD): Skew, Imb-A, and Imb-B. Imb-A and Imb-B represent imbalanced target distributions with different imbalance degrees, where Imb-A has $\rho = 100$ and Imb-B has $\rho = 50$.

| Dataset | Fashion-MNIST | | | CIFAR-10 | | | CINIC-10 | | | CIFAR-100 | | |
|---|---|---|---|---|---|---|---|---|---|---|---|---|
| TD | Skew | Imb-A | Imb-B | Skew | Imb-A | Imb-B | Skew | Imb-A | Imb-B | Skew | Imb-A | Imb-B |
| Local | 19.87 | 19.85 | 19.89 | 19.35 | 18.77 | 18.61 | 18.60 | 17.71 | 17.84 | 14.69 | 14.86 | 14.70 |
| FedAvg | 95.60 | 84.34 | 82.61 | 56.10 | 49.56 | 48.98 | 53.50 | 44.35 | 41.49 | 49.05 | 50.56 | 52.27 |
| FedProx | 89.55 | 78.01 | 76.64 | 42.20 | 40.72 | 45.22 | 50.80 | 41.49 | 33.17 | 50.40 | 52.45 | 52.76 |
| SCAFFOLD | 98.60 | 84.06 | 82.43 | 74.40 | 59.77 | 57.64 | 80.05 | 56.69 | 54.45 | 52.75 | 53.34 | 53.33 |
| FedDyn | 96.15 | 84.34 | 82.36 | 67.70 | 53.47 | 52.74 | 70.70 | 56.34 | 53.38 | 48.95 | 51.81 | 52.73 |
| MOON | 94.75 | 80.63 | 77.75 | 37.60 | 38.26 | 36.78 | 47.10 | 51.37 | 42.33 | 50.40 | 50.76 | 52.06 |
| FedDC | 93.00 | 81.84 | 81.79 | 52.15 | 47.42 | 52.23 | 84.40 | 55.13 | 52.38 | 39.95 | 47.47 | 50.31 |
| FedDecorr | 95.70 | 84.42 | 82.96 | 57.80 | 53.31 | 47.59 | 49.00 | 45.56 | 36.78 | 51.65 | 50.54 | 51.13 |
| FedAvgM | 93.00 | 82.53 | 82.04 | 55.00 | 50.65 | 51.88 | 48.60 | 40.84 | 38.60 | 49.10 | 50.26 | 50.24 |
| FedExP | 98.20 | 84.50 | 83.08 | 53.20 | 41.44 | 36.53 | 46.90 | 40.23 | 36.10 | 43.35 | 49.79 | 47.51 |
| FedDF | 92.05 | 77.72 | 67.41 | 68.95 | 42.37 | 42.03 | 68.95 | 47.70 | 45.65 | 22.90 | 19.90 | 26.91 |
| AFL | 95.40 | 86.68 | 84.36 | 62.90 | 56.70 | 54.07 | 50.10 | 43.38 | 40.50 | 54.10 | 54.01 | 53.30 |
| **FedSSA** | **99.50** | **86.97** | **84.69** | **90.00** | **62.99** | **58.43** | **92.50** | **61.30** | **55.78** | **78.05** | **58.13** | **53.77** |

of data samples that belong to category $c$, where $\rho$ denotes the imbalance ratio and $C$ is the total category number. The uniform distribution indicates a balanced test dataset.

**Training arguments.** We run FL for 100 communication rounds and train local models for $E_{local} = 10$ epochs with the batch size 64. We use SGD optimizer with a learning rate of 0.01. We utilize ResNet-18 (He et al., 2016) for CIFAR-100 and a simple CNN network with 3 convolutional layers and 3 fully-connected layers (Li et al., 2021; Collins et al., 2021) for others. Unless specified, the client numbers are 5 and 10 clients for NIID1 and NIID2, respectively. In FedSSA, we set $\lambda_{var} = 1$ and $\lambda_{we} = 1e - 3$. For all methods, hyper-parameters are tuned for each dataset.

## 5.2 RESULTS

**FL on data heterogeneity.** Considering two types of data heterogeneity on four datasets, we compare our proposed FedSSA with 12 representative baselines for three target distributions and report the results in Table 1 and Table 5 for NIID-1 and NIID-2, respectively. For each dataset, a skew distribution and two imbalanced distributions are considered.

From the table, we observe that: (1) FedSSA consistently achieves the best performance under all three target distributions across datasets and heterogeneity types, demonstrating its flexibility in adapting to different target distributions in FL. (2) Particularly for the skew target distribution, our FedSSA achieves significantly better results. For example, on the CIFAR-10 dataset, FedSSA outperforms others by 12.60% to 52.30% in NIID-1 and 8.56% to 22.31% in NIID-2. (3) Compared with FedDF (Lin et al., 2020) which also utilizes the unlabeled dataset but for knowledge distillation, FedSSA consistently performs better than baseline FedAvg (McMahan et al., 2017) while FedDF only outperforms FedAvg for some cases, indicating the effectiveness of the self-supervised learning scheme in dealing with the gap between training and target distributions.

**FL for sample-wise prediction.** We verify that FedSSA is still applicable when the unlabeled target dataset is unavailable during the FL training process. During the inference process, given a single sample, FedSSA automatically learns the weights for aggregating predictions, where the aggregated prediction is adopted as the final prediction, which is denoted as sample-wise FedSSA. We compare it with several ensemble baselines for fairness. Experiments are conducted under NIID-1 of CIFAR-10. Results in Table 2 show that sample-wise FedSSA consistently

Table 2: Sample-wise prediction task when data is only available after FL training process.

| Target Distribution | Skew | Imb. | Uni. |
|---|---|---|---|
| Local Training | 19.35 | 18.77 | 18.68 |
| Local Training + Ensemble | 38.50 | 39.43 | 48.68 |
| FedAvg | 37.05 | 39.96 | 42.28 |
| FedAvg + Ensemble | 69.70 | 50.12 | 49.73 |
| **Sample-wise FedSSA** | **72.50** | **50.66** | **50.37** |

outperforms other methods with the simple ensemble, indicating the broader effectiveness of self-supervised aggregation weight learning.

**Modularity of FedSSA.** One decent property of FedSSA is modularity, that is, our proposed FedSSA can be easily combined with many existing FL methods to further enhance the overall performance. Here, we consider three target distributions: skew, imbalanced ($\rho = 100$), and uniform under NIID-1 of CIFAR-10. We apply FedSSA on five representative algorithms that focus on local model training: FedAvg (McMahan et al., 2017), FedProx (Li et al., 2020b), SCAFFOLD (Karimireddy et al., 2020), FedDyn (Acar et al., 2020) and MOON (Li et al., 2021). For simplicity, we perform an additional ten epochs of aggregation weights learning based on their corresponding trained models at the last round. Results in Table 3 show that (1) in all target distribution settings, FedSSA consistently enhances the performance of these methods, indicating its plug-and-play property. (2) Specifically, for the most biased distribution (Skew), FedSSA brings 47.35% absolute accuracy improvement on average.

Table 3: Modularity of FedSSA under NIID-1 of CIFAR-10. We show the accuracy (Acc.) achieved by combining FedSSA with several baselines and the improvement ($\Delta$) brought by FedSSA. FedSSA consistently improves the baselines and brings significant improvement, especially for skew target distribution.

| **+FedSSA** | Skew | | Imbalanced | | Uniform | |
| --- | --- | --- | --- | --- | --- | --- |
| | Acc. | $\Delta$ | Acc. | $\Delta$ | Acc. | $\Delta$ |
| FedAvg | 96.55 | **+30.45** | 62.19 | **+16.23** | 47.24 | **+00.02** |
| FedProx | 93.90 | **+51.70** | 60.76 | **+20.04** | 39.24 | **+00.01** |
| SCAFFOLD | 94.95 | **+20.55** | 63.76 | **+03.99** | 47.59 | **+00.09** |
| FedDyn | 95.56 | **+27.86** | 61.62 | **+08.15** | 53.31 | **+00.07** |
| MOON | 95.85 | **+58.85** | 62.15 | **+23.89** | 56.85 | **+00.07** |

## 5.3 ANALYSIS OF FEDSSA

To provide more insights about FedSSA, we investigate the behaviors of FedSSA in detail by exploring the relationships among performance, local models, training distributions, and target distribution. Experiments are conducted under NIID-1 and skew target distribution on CIFAR-10. See Appendix D for more comparison experiments with some designed baselines.

**Performance analysis.** We evaluate the performances of each local model (denoted as L1 to L5), the global model obtained by FedAvg (McMahan et al., 2017) and FedSSA in Table 4. Results show that (1) L1 achieves the highest performance, indicating that more similar data distribution to the target may deserve a larger aggregation weight. (2) FedAvg achieves a moderate performance compared to L1 since its global model is obtained by equal aggregation. At the same time, L2 to L5 may bring a negative effect as they perform poorly on this target distribution. (3) FedSSA significantly outperforms FedAvg by 33.90% and achieves close to L1, indicating that FedSSA may assign a larger weight for the more representative model (L1).

Table 4: Performance analysis. The client 1's model L1 performs significantly better than L2 to L5. FedAvg achieves moderately as it equally aggregates all models while FedSSA performs much better as it adaptively adjusts the aggregation weights.

| Target | L1 | L2 | L3 | L4 | L5 | FedAvg | FedSSA |
| --- | --- | --- | --- | --- | --- | --- | --- |
| Skew | 97.30 | 0.50 | 0.20 | 0.40 | 0.50 | 56.10 | 90.00 |
| Imb. | 42.02 | 22.38 | 14.28 | 8.91 | 5.24 | 48.98 | 58.43 |

**Aggregation weight analysis.** Here, we show cosine similarity between target and clients' training distributions, and aggregation weights of FedAvg and FedSSA. As Figure 2 suggests, FedAvg uses dataset-size-based aggregation weights, failing to capture the inherent relationships between target and training distributions. On the contrary, FedSSA learns a set of aggregation weights that are more aligned to the cosine similarity relationships, indicating its adaptivity to target distribution.

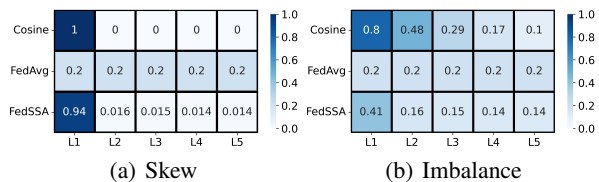

(a) Skew  (b) Imbalance

Figure 2: Aggregation weights analysis. FedAvg assigns equal aggregation weights while FedSSA assigns distinguishing aggregation weights, which are more aligned with the distribution similarity between local training distribution and target distribution.

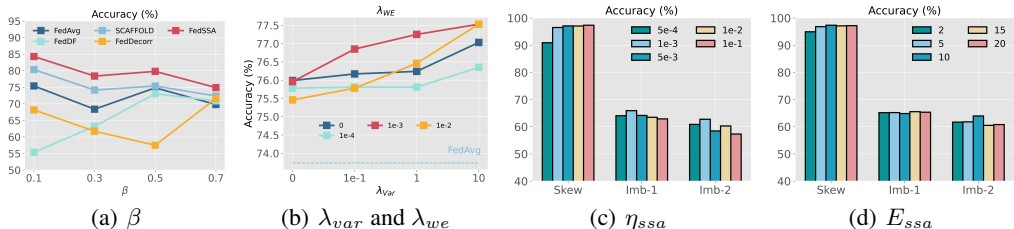

(a) $\beta$  (b) $\lambda_{var}$ and $\lambda_{we}$  (c) $\eta_{ssa}$  (d) $E_{ssa}$

Figure 4: (a) The effects of heterogeneity level under the Skew target distribution in NIID-1 setting on CIFAR-10. (b) Effects of $\lambda_{var}$ and $\lambda_{we}$ under imbalanced ($\rho = 50$) setting on CIFAR-10. (c) and (d) are ablation studies of different learning rates and the epochs for learning aggregation weights in FedSSA, respectively, under the NIID-1 setting on CIFAR-10.

## 5.4 ABLATION STUDY

Here, we investigate several key FL arguments: heterogeneity levels of training distributions and target distributions on CIFAR-10. Besides, we explore the effects of maximizing prediction confidence, maximizing weight entropy, learning rate, and epochs for learning aggregation weights in FedSSA. See Appendix C for experiments on the effects of the number of clients $K$ and local epochs.

**Different target distributions.** We evaluate the performance of FedSSA and FedAvg under NIID-1 and NIID-2 of CIFAR-10 with different target distributions: Skew, Imb-100 ($\rho = 100$), Imb-50, Imb-10, and Uniform (Uni.). Figure 3 shows that: (1) FedSSA consistently outperforms FedAvg, indicating its effectiveness in handling various target distributions. (2) The accuracy gap between FedSSA and FedAvg is especially large for Skew, where the gap between target and overall training distribution is also the largest. (3) They perform

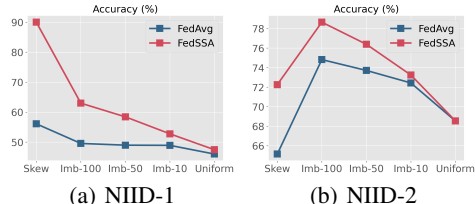

(a) NIID-1  (b) NIID-2

Figure 3: Performance on different target distributions under NIID-1 and NIID-2 settings.

similarly for uniform target distribution, which is reasonable since the target distribution is the same as the overall training distribution, thus the aggregation weights in FedAvg are appropriate.

**Different training distributions.** We tune the $\beta$ in NIID-2 (Hsu, Tzu-Ming Harry and Qi, Hang and Brown, Matthew, 2019) on CIFAR-10 (skew target distribution). As $\beta$ following $\{0.1, 0.3, 0.5, 0.7\}$ increases, data heterogeneity level decreases. Figure 4(a) shows that FedSSA outperforms the other 4 FL methods at various heterogeneity levels.

**Effects of variance- and entropy-related loss.** We tune $\lambda_{var}$ and $\lambda_{we}$ in equation 8 under the NIID-2 setting to validate the effectiveness of $\mathcal{L}_{var}$ and $\mathcal{L}_{we}$. Figure 4(b) shows that for a wide range, FedSSA always performs better than FedAvg, indicating the ease of hyper-parameters tuning. Generally, $\lambda_{var} = 10$ and $\lambda_{we} = 1e^{-3}$ performs better.

**Learning rate and epochs for learning aggregation weights.** We tune the learning rate $\eta_{ssa}$ and epochs $E_{ssa}$ under NIID-1. Results in Figure 4(c) and 4(d) show that FedSSA achieves stable performance with respect to the different learning rates and epochs.

## 6 CONCLUSIONS

Addressing the need to customize a global model tailored to unlabeled target data in Pragmatic Federated Learning, we propose FedSSA, a novel FL method with self-supervised aggregation. FedSSA learns aggregation weights that better fit the target distribution, by promoting transformation-invariant and confident predictions, and fostering collaborative ensemble. Extensive experiments show that FedSSA consistently outperforms state-of-the-art methods. Our work focuses on the severely unexplored gap between target and training distribution in FL. We hope that more future works can be proposed (e.g., a more general self-supervised strategy for diverse modalities) to tackle this practical issue and broaden the utility of FL.

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

# A   THEORETICAL ANALYSIS

We denote the global objective function as $F(\boldsymbol{\theta}) = \sum_{k=1}^{N} p_k F_k(\boldsymbol{\theta})$, where $\sum_{k=1}^{K} p_k = 1$, $F_k(\cdot)$ is the loss function of client $k$. Note that the general aggregation weight for local model aggregating is denoted by $w_k$ and the relative dataset size is denoted by $n_k = \frac{N_k}{\sum_i N_i}$. For FedAvg, the aggregation weight equals to relative dataset size: $w_k := n_k$, which optimizes the global objective function where $p_k := n_k$.

However, for an arbitrary target distribution, $p_k$ could be not strictly equal to $n_k$. For example, when the target distribution is exactly the same as Client 1 and distinct from others, the appropriate objective function should be $F(\boldsymbol{\theta}) = p_1 F_1(\boldsymbol{\theta})$, that is, $p_1 = 1.0$ and $p_k = 0.0$ for $k \neq 1$. In such cases, the optimal aggregation weight no longer follows $w_k = n_k$ but should be adaptively adjusted to align with the coefficient $p_k$ even though $p_k$ is not known in advance (since the target dataset is often unlabeled and the training distributions of clients are unknown due to privacy concerns).

In the following, we explore the effects of aggregation weight $w_k$ towards the optimization given a pre-defined global objective function (suppose $p_k$ is given).

**Assumption A.1 (Smoothness)** *Function $F_k(\mathbf{w})$ is Lipschitz-smooth: $||\nabla F_k(\mathbf{x}) - \nabla F_k(\mathbf{y})|| \leq L||\mathbf{x} - \mathbf{y}||$ for some $L$.*

**Assumption A.2 (Bounded Scalar)** *The global objective function $F(\mathbf{w})$ is bounded below by $F_{inf}$.*

**Assumption A.3 (Unbiased Gradient and Bounded Variance)** *For each client, the stochastic gradient is unbiased: $\mathbb{E}_\xi[g_k(\mathbf{w}|\xi)] = \nabla F_k(\mathbf{w})$, and has bounded variance: $\mathbb{E}_\xi[||g_k(\mathbf{w}|\xi) - \nabla F_k(\mathbf{w})||^2] \leq \sigma^2$.*

**Assumption A.4 (Bounded Dissimilarity)** *For each loss function $F_k(\mathbf{w})$, there exists constants $A, B > 0$ such that $||\nabla F_k(\mathbf{w})||^2 \leq A||\nabla F(\mathbf{w})||^2 + B$.*

Note that all assumptions are commonly used in federated learning literature (Wang et al., 2021c; Li et al., 2020b; 2019c; Reddi et al., 2020), except that we slightly modify the conventional bounded dissimilarity assumption for the process of theoretical derivation. Note that this modification does not affect the convergence rate. We present the optimization error bound in Theorem A.5 and leave the detailed proof in E.

**Theorem A.5 (Optimization bound of the global objective function)** *Let the global objective is $F(\mathbf{w}) = \sum_{k=1}^{K} p_k F_k(\mathbf{w})$. Under these Assumptions, if we set $\eta L \leq \frac{1}{2\tau}$, the optimization error will be bounded as follows:*

$$\min_t \mathbb{E} \left\| \nabla F(\boldsymbol{\theta}^{(t,0)}) \right\|^2 \leq \frac{1}{T} \sum_{t=0}^{T-1} \mathbb{E} \left\| \nabla F(\boldsymbol{\theta}^{(t,0)}) \right\|^2$$

$$\leq \frac{1}{1 - C - 2AC - W_D(1-C)}$$
$$\left( \underbrace{\frac{2(1-C)\left(F(\boldsymbol{\theta}^{(0,0)}) - F_{inf}\right)}{\tau \eta T}}_{T_1} + \underbrace{(1-C)BW_D}_{T_2} \right.$$
$$\left. + \underbrace{2(1-C)L\eta\sigma^2 \sum_{k=1}^{K} w_k^2}_{T_3} + \underbrace{2(\tau-1)\sigma^2 L^2 \eta^2}_{T_4} + \underbrace{2BC}_{T_5} \right),$$

*where $W_D = 2K\left[\sum_{k=1}^{K}(p_k - w_k)^2\right]$, $w_k$ is the aggregation weight, $p_k$ is the coefficient of global objective function, $C = 2\tau(\tau - 1)\eta^2 L^2 < 1$, $\tau$ is the number of steps in local model training, $\eta$ is learning rate, $T$ is the total communication round in FL, $K$ is the total client number, $F_{inf}, A, B, L, \sigma$ are the constants in assumptions.*

From the theorem A.5, we have the following two observations about the effects of the aggregation weights towards the optimization upper bound.

**Remark A.6 (Optimal Aggregation)** *When the aggregation weight $w_k$ strictly equals the coefficient $p_k$ in the global objective function, the $W_D \to 0$. As a result, the denominator in the upper bound becomes the largest and the term $T_2$ vanishes, indicating that this upper bound will be the tightest.*

**Remark A.7 (Aggregation based on FedAvg)** *As FedAvg applies dataset-size-based aggregation where $w_k := n_k$, there are often cases where $p_k \neq n_k$ due to the arbitrariness of the target distribution. This results in a $W_D > 0$ and makes the upper bound looser.*

Thus, there exists an optimal set of aggregation weights that minimizes the optimization bound of the global objective function while the traditional aggregation weights applied by most FL methods could be far from optimality for cases of different target distributions. **This motivates an adaptive algorithm that automatically adjusts the aggregation weights according to the target distribution and determines a set of aggregation weights that follow $w_k \approx p_k$.**

When the aggregation weights are optimal $w_k := p_k$, we have the following corollary.

**Corollary A.8 (Convergence rate for optimal aggregation weights)** *Given $W_D = 0$, by setting $\eta = \frac{1}{\sqrt{\tau T}}$, the optimization bound in Theorem A.5 can be re-written as*

$$\min_t \mathbb{E} \left\| \nabla F(\boldsymbol{\theta}^{(t,0)}) \right\|^2 \leq \mathcal{O}(\frac{1}{\sqrt{\tau T}}) + \mathcal{O}(\frac{1}{T}) + \mathcal{O}(\frac{\tau}{T}).$$

From the corollary, we observe that the convergence rate aligns with most FL theoretical literature (Li et al., 2019b; Wang et al., 2021c;b), verifying the correctness of the above theoretical results.

## B IMPLEMENTATION

### B.1 ALGORITHM FOR FEDSSA

Here, we show the Algorithm 1 of the proposed FedSSA. Specifically, compared with vanilla FedAvg (McMahan et al., 2017), our algorithm adds the flexible aggregation-weights learning module.

---

**Algorithm 1** FedSSA

**Input:** Total round $T$, initial global model $\boldsymbol{\theta}^0$, $K$ clients with training dataset $\mathcal{D}_k$.
**Output:** Global model.
**for** $t = 0, 1, ..., T - 1$ **do**
    Server sends global model $\boldsymbol{\theta}^t$ to initialize each client
    // Local Model Training
    **for** $k = 0, 1, ..., K - 1$ in parallel **do**
        $\boldsymbol{\theta}_k^t := \arg\min_{\boldsymbol{\theta}_k} \mathcal{L}_c (\boldsymbol{\theta}_k; \mathcal{D}_k)$
        Sends local model $\boldsymbol{\theta}_k^t$ to the server
    **end for**
    // Aggregation Weights Optimizing
    $\{w_k^t\} := \arg\min_{\{w_k\}} \mathcal{L}_{SSA} (\{\boldsymbol{\theta}_k^t\}, \{w_k\}; \mathcal{U})$
    // Global Model Aggregation
    $\boldsymbol{\theta}^{t+1} := \sum_k w_k^t \boldsymbol{\theta}_k^t$
**end for**
**return** final global model $\boldsymbol{\theta}^T$

---

### B.2 BASELINES AND HYPER-PARAMETERS

- Local training: each local model is individually trained under each local dataset and there is no model interaction. Each model is evaluated on the target dataset and we report the averaged result.

Table 5: Classification accuracy (%) comparison under NIID-2 on Fashion-MNIST, CIFAR-10, CINIC, and CIFAR-100, considering 3 target distributions (TD): Skew, Imb-A, Imb-B. Imb-A and Imb-B denote imbalanced target distribution with two different imbalance degrees, where Imb-A with $\rho = 100$ and Imb-B with $\rho = 50$.

| Dataset | Fashion-MNIST | | | CIFAR-10 | | | CINIC-10 | | | CIFAR-100 | | |
|---|---|---|---|---|---|---|---|---|---|---|---|---|
| TD | Skew | Imb-A | Imb-B | Skew | Imb-A | Imb-B | Skew | Imb-A | Imb-B | Skew | Imb-A | Imb-B |
| Local | 68.88 | 71.08 | 70.48 | 37.58 | 41.72 | 41.15 | 27.34 | 44.06 | 42.21 | 21.87 | 20.95 | 21.42 |
| FedAvg | 85.87 | 88.30 | 88.62 | 65.15 | 74.82 | 73.74 | 35.30 | 68.32 | 65.55 | 53.62 | 54.34 | 54.59 |
| FedProx | 76.93 | 81.15 | 81.04 | 64.22 | 72.24 | 71.81 | 44.33 | 64.53 | 63.54 | 52.48 | 53.03 | 54.30 |
| SCAFFOLD | 87.63 | 90.27 | 89.73 | 67.87 | 75.38 | 74.35 | 36.73 | 60.86 | 65.01 | 59.73 | 59.83 | 59.98 |
| FedDyn | 86.06 | 87.69 | 88.62 | 71.23 | 73.20 | 72.38 | 29.12 | 67.15 | 66.12 | 57.66 | 58.24 | 57.92 |
| MOON | 85.47 | 88.05 | 87.84 | 63.70 | 76.03 | 74.24 | 29.39 | 68.56 | 66.19 | 52.02 | 53.92 | 54.59 |
| FedDC | 86.50 | 88.10 | 88.16 | 69.36 | 74.37 | 75.24 | 40.46 | 65.01 | 62.90 | 57.14 | 58.85 | 58.09 |
| FedDecorr | 92.80 | 87.73 | 88.37 | 57.48 | 74.78 | 75.09 | 25.83 | 66.63 | 65.37 | 56.16 | 56.08 | 55.79 |
| FedAvgM | 92.42 | 88.46 | 88.26 | 67.77 | 76.43 | 74.74 | 33.79 | 65.94 | 65.12 | 48.76 | 50.54 | 51.29 |
| FedExP | 93.35 | 89.87 | 89.55 | 63.42 | 71.07 | 72.95 | 27.92 | 41.28 | 56.74 | 39.75 | 36.13 | 36.36 |
| FedDF | 93.36 | 88.58 | 88.05 | 61.55 | 73.04 | 70.16 | 23.34 | 66.22 | 64.69 | 22.57 | 21.82 | 21.92 |
| AFL | 85.25 | 87.97 | 87.76 | 66.04 | 77.20 | 76.31 | 37.17 | 67.68 | 66.37 | 53.99 | 53.59 | 55.03 |
| **FedSSA** | **96.40** | **90.92** | **90.20** | **79.79** | **79.58** | **77.39** | **49.62** | **70.34** | **66.51** | **63.20** | **68.32** | **60.78** |

- FedProx (Li et al., 2020b): it applies $\ell_2$ distance regularization and the hyper-parameter used is $0.01$.

- FedAvgM (Hsu et al., 2019): it applies momentum at the server side and the hyper-parameter used is $0.7$.

- SCAFFOLD (Karimireddy et al., 2020): it applies a control variate to correct the local gradient.

- FedDF (Lin et al., 2020): it distills the knowledge of local models to the global model by using the unlabeled target dataset.

- FedDyn (Acar et al., 2020): it introduces a dynamic regularization term and the hyper-parameter used is $0.01$.

- MOON (Li et al., 2021): it conducts contrastive learning among the current local model, previous global model (positive), and previous local model (negative). The hyper-parameter used is $0.1$.

- FedDC (Gao et al., 2022b): it introduces an auxiliary local drift variable and the hyper-parameter used is $0.01$.

- FedExP (Jhunjhunwala et al., 2023): it dynamically adjusts the global learning rate with the hyper-parameter $0.001$.

- FedDecorr (Shi et al., 2022): it regularizes feature correlation during local model training and the hyper-parameter used is $0.1$.

- AFL (Mohri et al., 2019): it computes stochastic gradients with respect to the weights and updates the model accordingly, then conducts a projection step with hyper-parameter $0.001$.

## B.3 MODELS

The simple CNN network sequentially consists of $5 \times 5$ convolution layer, max-pooling layer, $5 \times 5$ convolution layer, and three fully-connected layers with the hidden size of $120$, $84$, and $10$ respectively. We use ResNet18 (He et al., 2016) in the Pytorch library. We replace the first $7 \times 7$ convolution layer with a $3 \times 3$ convolution layer and eliminate the first pooling layer.

Table 6: Effects of number of clients $K$ under skew target distribution on NIID-2 of CIFAR-10. The participation rate is 0.2 for $K = 50$ and 1.0 for others.

| $K$ | 5 | 10 | 20 | 30 | 50 |
|---|---|---|---|---|---|
| FedAvg (McMahan et al., 2017) | 65.10 | 65.15 | 77.94 | 69.21 | 79.83 |
| FedDF (Lin et al., 2020) | 62.31 | 61.55 | 65.17 | 59.52 | 76.98 |
| SCAFFOLD (Karimireddy et al., 2020) | 69.30 | 67.87 | 79.92 | 70.84 | 79.10 |
| FedDecorr (Shi et al., 2022) | 57.94 | 57.48 | 75.30 | 61.63 | 82.88 |
| **FedSSA (ours)** | **81.78** | **79.79** | **81.46** | **72.01** | **84.45** |

## C  EXPERIMENTS

### C.1  BASIC EXPERIMENT RESULTS FOR NIID-2

We conduct similar experiments on the fashion-MNIST (Xiao et al., 2017), CIFAR-10 (Krizhevsky et al., 2009), CINIC-10 (Darlow et al., 2018) and CIFAR-100 (Krizhevsky et al., 2009) datasets, under the NIID-2 setting, to complement Table 1 in the main content. As before, we compare our results with 12 existing baselines. Here, we choose to apply FedSSA on well-performed baselines and thus we apply FedSSA on MOON and SCAFFOLD for CINIC-10 and CIFAR-100, respectively. Results in Table 5 show that (1) FedSSA consistently outperforms the baselines with a large margin on average. For example, on the CIFAR-10 dataset, FedSSA outperforms others by 12.60% to 52.30% in NIID-1 and 8.56% to 22.31% in NIID-2. (2) Tuning the aggregation weights according to our FedSSA algorithm leads to significant performance improvements on average for a well-performed baseline, indicating the plug-and-play property of FedSSA.

### C.2  EFFECTS OF NUMBER OF CLIENTS

We tune the number of clients $K$ in $\{5, 10, 20, 30, 50\}$ under skew target distribution on NIID-1 of CIFAR-10. When $K = 50$, we consider a 0.2 participating rate. Results in Table 6 show that (1) FedSSA performs the best across different client numbers, and (2) FedSSA shows applicability toward partial client participation scenarios.

### C.3  EFFECTS OF LOCAL EPOCHS

Here, we explore the impact of the number of local epochs $E_{local}$ during local model training. Experiments are conducted on the skew target distribution under both NIID-1 and NIID-2 settings on CIFAR-10. Specifically, we tune the number of local epochs $E_{local}$ in $\{2, 5, 10, 20\}$ and compare the results with five representative FL methods, including FedAvg (McMahan et al., 2017), SCAFFOLD (Karimireddy et al., 2020), FedDF (Lin et al., 2020) and FedDecorr (Shi et al., 2022). As illustrated in Figure 5, the performance fluctuations of FedSSA were within 2%, whereas other methods exhibited fluctuations ranging from 75.90% to 46.80%. These results demonstrate that FedSSA is less sensitive to the number of local epochs compared to other FL methods, exhibiting its stability.

### C.4  EFFECTS OF VARIANCE- AND ENTROPY-RELATED LOSS

Here, we complement the results in Figure 5 (b) in the main content. We tune $\lambda_{var}$ in $\{0, 1e - 1, 1, 10\}$ and $\lambda_{we}$ in $\{0, 1e - 4, 1e - 3, 1e - 2\}$, respectively. We show the results under another two settings in Table 7 and Table 8. We observe that: (1) in NIID-1 setting, $\lambda_{var}$ and $\lambda_{we}$ have relatively ordinary influence on the performance of FedSSA; (2) in the NIID-2 setting, $\lambda_{var}$ brings about 10% improvement regardless of $\lambda_{we}$, indicating the confidence-promoting loss term enhance significantly the performance of the aggregated model.

These two results show that when the computing resource is adequate, we can tune both two hyperparameters for better performance. However, for cases where the computing resource is limited, we can only apply the transformation-variant loss term.

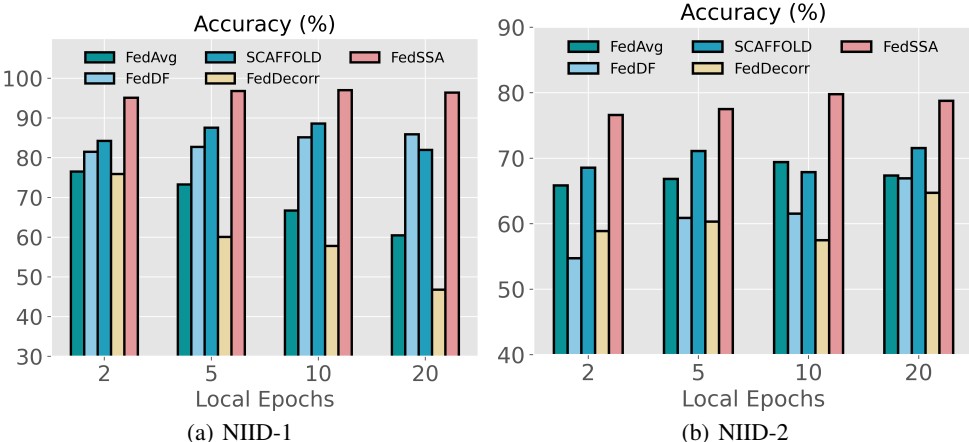

Figure 5: Effects of local epochs on CIFAR-10 under NIID-1 and NIID-2 settings.

Table 7: Effects of hyper-parameters for weight-entropy loss and confidence-based loss term under imbalanced ($\rho = 100$) target distribution on NIID-1 of CIFAR-10. Note that $\lambda_{we} = \lambda_{var} = 0.0$ denotes FedSSA with our proposed transformation-invariant loss only.

| Accuracy | $\lambda_{var}$ | | | |
|---|---|---|---|---|
| $\lambda_{we}$ | 0 | 1e-1 | 1 | 10 |
| 0 | 65.05 | 64.65 | 64.85 | 63.20 |
| 1e-4 | 64.49 | 64.41 | 64.65 | 63.60 |
| 1e-3 | 65.01 | 64.73 | 65.05 | 65.42 |
| 1e-2 | 64.85 | 65.86 | 65.01 | 64.53 |

## D  SIMPLE DESIGNED BASELINES

Based on the transformation-invariant and confidence-promoting principles, we design two straightforward algorithms to adjust the aggregation weights based on pre-defined rules.

**Transformation-invariant based weights.** For each local model $\boldsymbol{\theta}_k$, we can obtain the predictions $\hat{\boldsymbol{y}}_{k,m}^{(1,2)}$ of two transformation views of the same sample $\boldsymbol{x}_m$. Then, we can calculate the cosine similarity between these two predictions for each data sample. The averaged cosine similarity for each client $k$ is denoted as $s_k$

$$s_k = \frac{1}{N_k} \sum_{m=1}^{N_k} \cos\left(\hat{\boldsymbol{y}}_{k,m}^{(1)}, \hat{\boldsymbol{y}}_{k,m}^{(2)}\right). \tag{9}$$

Finally, we normalize the cosine similarity as the aggregation weight for each client: $w_k = \frac{s_k}{\sum_i s_i}$.

**Confidence-promoting based weights.** Similarly, we first calculate the prediction confidence for each client $k$, denoted as $c_k$:

$$c_k = \frac{1}{N_k} \sum_{m=1}^{N_k} var(\hat{\boldsymbol{y}}_{k,m}). \tag{10}$$

Aggregation weights are obtained by normalization: $w_k = \frac{c_k}{\sum_i c_i}$.

**Experiments.** Considering the NIID-1 and NIID-2 settings, we conduct experiments on CIFAR-10, shown in Table 9. Results show that (1) these proposed simple baselines and FedSSA outperforms FedAvg across different settings, indicating that both the similarity-based and confidence-based metrics are beneficial for determining aggregation weights. (2) Our learnable aggregation weights based on enhancing both transformation-invariance and confidence contribute to the best results, indicating the effectiveness of FedSSA.

Table 8: Effects of hyper-parameters for weight-entropy loss and confidence-based loss term under skew target distribution on NIID-2 of CIFAR-10. Note that $\lambda_{we} = \lambda_{var} = 0.0$ denotes FedSSA with our proposed transformation-invariant loss only.

| Accuracy | $\lambda_{var}$ | | | |
|---|---|---|---|---|
| $\lambda_{we}$ | 0 | 1e-1 | 1 | 10 |
| 0 | 70.81 | 71.14 | 70.95 | 80.50 |
| 1e-4 | 70.29 | 69.93 | 69.78 | 79.75 |
| 1e-3 | 70.63 | 71.66 | 71.56 | 79.84 |
| 1e-2 | 71.52 | 70.53 | 69.87 | 80.87 |

Table 9: Comparisons with designed baselines on CIFAR-10.

| Setting | NIID-1 | | | NIID-2 | | |
|---|---|---|---|---|---|---|
| Target Distribution | Skew | Imb-A | Imb-B | Skew | Imb-A | Imb-B |
| FedAvg | 56.10 | 49.56 | 48.98 | 65.15 | 74.82 | 73.74 |
| Similarity | 72.50 | 53.03 | 52.27 | 68.52 | 76.43 | 75.13 |
| Confidence | 70.00 | 58.79 | 52.16 | 70.72 | 74.33 | 74.49 |
| **FedSSA** | **90.00** | **62.99** | **58.43** | **79.79** | **79.58** | **77.39** |

# E PROOF

## E.1 PRELIMINARIES

The global objective function is $F(\boldsymbol{\theta}) = \sum_{k=1}^{N} p_k F_k(\boldsymbol{\theta})$, where $\sum_{k=1}^{K} p_k = 1$. Note that the general aggregation weight for local model aggregating is denoted by $w_k$ and the relative dataset size is denoted by $n_k = \frac{N_k}{\sum_i N_i}$. For FedAvg, the aggregation weight equals to relative dataset size: $w_k := n_k$, which optimizes the global objective function where $p_k := n_k$. For an arbitrary target distribution, $p_k$ should not be strictly equal to $n_k$ like FedAvg and so do $w_k$. For example, when the target distribution is exactly the same as Client 1 and distinct from others, the objective function should be $F(\boldsymbol{\theta}) = p_1 F_1(\boldsymbol{\theta})$, that is, $p_1 = 1.0$ and $p_k = 0.0$ for $k \neq 1$. Thus, we explore the effects of aggregation weight $w_k$ towards the optimization given a pre-defined global objective function, that is, $p_k$ is given.

For ease of writing, we use $g_k(\boldsymbol{\theta})$ to denote mini-batch gradient $g_k(\boldsymbol{\theta}|\xi)$ and $\nabla F_k(\boldsymbol{\theta})$ to denote full-batch gradient, where $\xi$ is a mini-batch sampled from dataset. We further define the following two notions:

$$\text{Averaged Mini-batch Gradient:} \quad \mathbf{d}_k = \frac{1}{\tau} \sum_{r=0}^{\tau-1} g_k(\boldsymbol{\theta}_k^{(t,r)}), \tag{11}$$

$$\text{Averaged Full-batch Gradient:} \quad \mathbf{h}_k = \frac{1}{\tau} \sum_{r=0}^{\tau-1} \nabla F_k(\boldsymbol{\theta}_k^{(t,r)}). \tag{12}$$

Then, the update of the global model between the two rounds is as follows:

$$\boldsymbol{\theta}^{(t+1,0)} - \boldsymbol{\theta}^{(t,0)} = -\tau\eta \sum_{k=1}^{K} w_k \mathbf{d}_k. \tag{13}$$

Here, we present a key lemma and defer its proof to section E.3.

### E.1.1 LEMMA 1 (WANG ET AL., 2021C).

*Suppose $\{A_t\}_{t=1}^T$ is a sequence of random matrices and follows $\mathbb{E}[A_t|A_{t-1}, A_{t-2}, ..., A_1] = \mathbf{0}$, then*

$$\mathbb{E}\left[\left\|\sum_{t=1}^T A_t\right\|_F^2\right] = \sum_{t=1}^T \mathbb{E}\left[\|A_t\|_F^2\right]$$

### E.2 PROOF OF THEOREM 1

According to the Lipschitz-smooth assumption in Assumption 1, we have its equivalent form (Bottou et al., 2018)

$$
\begin{aligned}
&\mathbb{E}\left[F(\boldsymbol{\theta}^{(t+1,0)})\right] - F(\boldsymbol{\theta}^{(t,0)}) \\
&\leq \mathbb{E}\left[\left\langle \nabla F(\boldsymbol{\theta}^{(t,0)}), \boldsymbol{\theta}^{(t+1,0)} - \boldsymbol{\theta}^{(t,0)}\right\rangle\right] - \frac{L}{2}\mathbb{E}\left[\left\|\boldsymbol{\theta}^{(t+1,0)} - \boldsymbol{\theta}^{(t,0)}\right\|^2\right] \quad (14) \\
&= -\tau\eta\underbrace{\mathbb{E}\left[\left\langle \nabla F(\boldsymbol{\theta}^{(t,0)}), \sum_{k=1}^K w_k\mathbf{d}_k\right\rangle\right]}_{N_1} + \frac{L\tau^2\eta^2}{2}\underbrace{\mathbb{E}\left[\left\|\sum_{k=1}^K w_k\mathbf{d}_k\right\|^2\right]}_{N_2}, \quad (15)
\end{aligned}
$$

where the expectation is taken over mini-batches $\xi_k^{(t,r)}$, $\forall k \in 1, 2, ..., K$, $r \in 0, 1, ..., \tau - 1$.

### E.2.1 BOUNDING $N_1$ IN (15)

$$
\begin{aligned}
N_1 &= \mathbb{E}\left[\left\langle \nabla F(\boldsymbol{\theta}^{(t,0)}), \sum_{k=1}^K w_k(\mathbf{d}_k - \mathbf{h}_k)\right\rangle\right] + \mathbb{E}\left[\left\langle \nabla F(\boldsymbol{\theta}^{(t,0)}), \sum_{k=1}^K w_k\mathbf{h}_k\right\rangle\right] \quad (16) \\
&= \mathbb{E}\left[\left\langle \nabla F(\boldsymbol{\theta}^{(t,0)}), \sum_{k=1}^K w_k\mathbf{h}_k\right\rangle\right] \quad (17) \\
&= \frac{1}{2}\left\|\nabla F(\boldsymbol{\theta}^{(t,0)})\right\|^2 + \frac{1}{2}\mathbb{E}\left[\left\|\sum_{k=1}^K w_k\mathbf{h}_k\right\|^2\right] - \frac{1}{2}\mathbb{E}\left[\left\|\nabla F(\boldsymbol{\theta}^{(t,0)}) - \sum_{k=1}^K w_k\mathbf{h}_k\right\|^2\right], \quad (18)
\end{aligned}
$$

where (17) uses the unbiased gradient assumption in Assumption 3, such that $\mathbb{E}[\mathbf{d}_k - \mathbf{h}_k] = \mathbf{h}_k - \mathbf{h}_k = \mathbf{0}$. (18) uses the fact that $2\langle a, b\rangle = \|a\|^2 + \|b\|^2 - \|a - b\|^2$.

### E.2.2 Bounding $N_2$ in (15)

$$N_2 = \mathbb{E}\left[\left\|\sum_{k=1}^{K} w_k(\mathbf{d}_k - \mathbf{h}_k) + \sum_{k=1}^{K} w_k \mathbf{h}_k\right\|^2\right] \tag{19}$$

$$\leq 2\mathbb{E}\left[\left\|\sum_{k=1}^{K} w_k(\mathbf{d}_k - \mathbf{h}_k)\right\|^2\right] + 2\mathbb{E}\left[\left\|\sum_{k=1}^{K} w_k \mathbf{h}_k\right\|^2\right] \tag{20}$$

$$= 2\sum_{k=1}^{K} w_k^2 \mathbb{E}\left[\|\mathbf{d}_k - \mathbf{h}_k\|^2\right] + 2\mathbb{E}\left[\left\|\sum_{k=1}^{K} w_k \mathbf{h}_k\right\|^2\right] \tag{21}$$

$$= \frac{2}{\tau^2}\sum_{k=1}^{K} w_k^2 \mathbb{E}\left[\left\|\sum_{r=0}^{\tau-1}(g_k(\boldsymbol{\theta}_k^{(t,r)}) - \nabla F_k(\boldsymbol{\theta}_k^{(t,r)}))\right\|^2\right] + 2\mathbb{E}\left[\left\|\sum_{k=1}^{K} w_k \mathbf{h}_k\right\|^2\right] \tag{22}$$

$$= \frac{2}{\tau^2}\sum_{k=1}^{K} w_k^2 \sum_{r=0}^{\tau-1} \mathbb{E}\left[\left\|g_k(\boldsymbol{\theta}_k^{(t,r)}) - \nabla F_k(\boldsymbol{\theta}_k^{(t,r)})\right\|^2\right] + 2\mathbb{E}\left[\left\|\sum_{k=1}^{K} w_k \mathbf{h}_k\right\|^2\right] \tag{23}$$

$$\leq \frac{2\sigma^2}{\tau}\sum_{k=1}^{K} w_k^2 + 2\mathbb{E}\left[\left\|\sum_{k=1}^{K} w_k \mathbf{h}_k\right\|^2\right] \tag{24}$$

where (20) follows $\|a + b\|^2 \leq 2\|a\|^2 + 2\|b\|^2$, (21) uses the fact that clients are independent to each other so that $\mathbb{E}\langle \mathbf{d}_k - \mathbf{h}_k, \mathbf{d}_n - \mathbf{h}_n\rangle = 0, \forall k \neq n$. (23) uses Lemma 1 and (24) uses bounded variance assumption in Assumption 2.

Plug (18) and (24) back into (15), we have

$$\mathbb{E}\left[F(\boldsymbol{\theta}^{(t+1,0)})\right] - F(\boldsymbol{\theta}^{(t,0)})$$

$$\leq -\frac{\tau\eta}{2}\left\|\nabla F(\boldsymbol{\theta}^{(t,0)})\right\|^2 - \frac{\tau\eta}{2}(1 - 2\tau\eta L)\mathbb{E}\left[\left\|\sum_{k=1}^{K} w_k \mathbf{h}_k\right\|^2\right] + L\tau\eta^2\sigma^2\sum_{k=1}^{K} w_k^2 + \frac{\tau\eta}{2}\underbrace{\mathbb{E}\left[\left\|\nabla F(\boldsymbol{\theta}^{(t,0)}) - \sum_{k=1}^{K} w_k \mathbf{h}_k\right\|^2\right]}_{N_3}.$$

$$\tag{25}$$

### E.2.3 Bounding $N_3$ in (25)

$$\mathbb{E}\left[\left\|\nabla F(\boldsymbol{\theta}^{(t,0)}) - \sum_{k=1}^{K} w_k \mathbf{h}_k\right\|^2\right]$$

$$= \mathbb{E}\left[\left\|\sum_{k=1}^{K}(p_k - w_k)\nabla F_k(\boldsymbol{\theta}^{(t,0)}) + \sum_{k=1}^{K} w_k\left(\nabla F_k(\boldsymbol{\theta}^{(t,0)}) - \mathbf{h}_k\right)\right\|^2\right] \tag{26}$$

$$\leq 2\left\|\sum_{k=1}^{K}(p_k - w_k)\nabla F_k(\boldsymbol{\theta}^{(t,0)})\right\|^2 + 2\left\|\sum_{k=1}^{K} w_k\left(\nabla F_k(\boldsymbol{\theta}^{(t,0)}) - \mathbf{h}_k\right)\right\|^2 \tag{27}$$

$$\leq 2\left[\sum_{k=1}^{K}(p_k - w_k)^2\right]\left[\sum_{k=1}^{K}\left\|\nabla F_k(\boldsymbol{\theta}^{(t,0)})\right\|^2\right] + 2\left\|\sum_{k=1}^{K} w_k\left(\nabla F_k(\boldsymbol{\theta}^{(t,0)}) - \mathbf{h}_k\right)\right\|^2 \tag{28}$$

$$\leq 2K\left[\sum_{k=1}^{K}(p_k - w_k)^2\right]\left[A\left\|\nabla F(\boldsymbol{\theta}^{(t,0)})\right\|^2 + B\right] + 2\left\|\sum_{k=1}^{K} w_k\left(\nabla F_k(\boldsymbol{\theta}^{(t,0)}) - \mathbf{h}_k\right)\right\|^2, \tag{29}$$

where (27) follows $\|a + b\|^2 \leq 2\|a\|^2 + 2\|b\|^2$, (28) follows Cauchy–Schwarz inequality, (29) uses the bounded similarity assumption in Assumption 4.

We use $W_D$ to denote $2K\left[\sum_{k=1}^{K}(p_k - w_k)^2\right]$. When $1 - 2\tau\eta L \geq 0$, we have

$$
\mathbb{E}\left[F(\boldsymbol{\theta}^{(t+1,0)})\right] - F(\boldsymbol{\theta}^{(t,0)})
$$

$$
\leq -\frac{\tau\eta(1 - AW_D)}{2}\left\|\nabla F(\boldsymbol{\theta}^{(t,0)})\right\|^2 + L\tau\eta^2\sigma^2\sum_{k=1}^{K}w_k^2 + \frac{\tau\eta BW_D}{2} + \tau\eta\mathbb{E}\left[\left\|\sum_{k=1}^{K}w_k\left(\nabla F_k(\boldsymbol{\theta}^{(t,0)}) - \mathbf{h}_k\right)\right\|^2\right]
$$

$$
\tag{30}
$$

$$
\leq -\frac{\tau\eta(1 - AW_D)}{2}\left\|\nabla F(\boldsymbol{\theta}^{(t,0)})\right\|^2 + L\tau\eta^2\sigma^2\sum_{k=1}^{K}w_k^2 + \frac{\tau\eta BW_D}{2} + \tau\eta\sum_{k=1}^{K}w_k\underbrace{\mathbb{E}\left[\left\|\nabla F_k(\boldsymbol{\theta}^{(t,0)}) - \mathbf{h}_k\right\|^2\right]}_{N_4},
$$

$$
\tag{31}
$$

where (31) uses Jensen's Inequality $\left\|\sum_{k=1}^{K}w_k x_k\right\|^2 \leq \sum_{k=1}^{K}w_k\|x_k\|^2$.

### E.2.4 BOUNDING $N_4$ IN (31)

$$
\mathbb{E}\left[\left\|\nabla F_k(\boldsymbol{\theta}^{(t,0)}) - \mathbf{h}_k\right\|^2\right] = \mathbb{E}\left[\left\|\nabla F_k(\boldsymbol{\theta}^{(t,0)}) - \frac{1}{\tau}\sum_{r=0}^{\tau-1}\nabla F_k(\boldsymbol{\theta}_k^{(t,r)})\right\|^2\right] \tag{32}
$$

$$
= \mathbb{E}\left[\left\|\frac{1}{\tau}\sum_{r=0}^{\tau-1}(\nabla F_k(\boldsymbol{\theta}^{(t,0)}) - \nabla F_k(\boldsymbol{\theta}_k^{(t,r)}))\right\|^2\right] \tag{33}
$$

$$
\leq \frac{1}{\tau}\sum_{r=0}^{\tau-1}\mathbb{E}\left[\left\|\nabla F_k(\boldsymbol{\theta}^{(t,0)}) - \nabla F_k(\boldsymbol{\theta}_k^{(t,r)})\right\|^2\right] \tag{34}
$$

$$
\leq \frac{L^2}{\tau}\sum_{r=0}^{\tau-1}\underbrace{\mathbb{E}\left[\left\|\boldsymbol{\theta}^{(t,0)} - \boldsymbol{\theta}_k^{(t,r)}\right\|^2\right]}_{N_5}, \tag{35}
$$

where (34) uses Jensen's Inequality and (35) follows Lipschitz-smooth property.

### E.2.5  BOUNDING $N_5$ IN (41)

$$\mathbb{E}\left[\left\|\boldsymbol{\theta}^{(t,0)} - \boldsymbol{\theta}_k^{(t,r)}\right\|^2\right] = \eta^2 \mathbb{E}\left[\left\|\sum_{s=0}^{r-1} g_k(\boldsymbol{\theta}_k^{(t,s)})\right\|^2\right] \tag{36}$$

$$\leq 2\eta^2 \mathbb{E}\left[\left\|\sum_{s=0}^{r-1}\left(g_k(\boldsymbol{\theta}_k^{(t,s)}) - \nabla F_k(\boldsymbol{\theta}_k^{(t,s)})\right)\right\|^2\right] + 2\eta^2 \mathbb{E}\left[\left\|\sum_{s=0}^{r-1} \nabla F_k(\boldsymbol{\theta}_k^{(t,s)})\right\|^2\right] \tag{37}$$

$$= 2\eta^2 \sum_{s=0}^{r-1} \mathbb{E}\left[\left\|g_k(\boldsymbol{\theta}_k^{(t,s)}) - \nabla F_k(\boldsymbol{\theta}_k^{(t,s)})\right\|^2\right] + 2\eta^2 \mathbb{E}\left[\left\|\sum_{s=0}^{r-1} \nabla F_k(\boldsymbol{\theta}_k^{(t,s)})\right\|^2\right] \tag{38}$$

$$\leq 2r\eta^2\sigma^2 + 2\eta^2 \mathbb{E}\left[\left\|r\sum_{s=0}^{r-1}\frac{1}{r} \nabla F_k(\boldsymbol{\theta}_k^{(t,s)})\right\|^2\right] \tag{39}$$

$$\leq 2r\eta^2\sigma^2 + 2r\eta^2 \sum_{s=0}^{r-1} \mathbb{E}\left[\left\|\nabla F_k(\boldsymbol{\theta}_k^{(t,s)})\right\|^2\right] \tag{40}$$

$$\leq 2r\eta^2\sigma^2 + 2r\eta^2 \sum_{s=0}^{\tau-1} \mathbb{E}\left[\left\|\nabla F_k(\boldsymbol{\theta}_k^{(t,s)})\right\|^2\right] \tag{41}$$

where (37) uses $\|a + b\|^2 \leq 2\|a\|^2 + 2\|b\|^2$, (38) uses Lemma 1, (39) uses the bounded variance assumption in Assumption 3, (40) uses Jensen's Inequality.

Plug (41) back into (35) and use this equation $\sum_{r=0}^{\tau-1} r = \frac{\tau(\tau-1)}{2}$, we have

$$\mathbb{E}\left[\left\|\nabla F_k(\boldsymbol{\theta}^{(t,0)}) - \mathbf{h}_k\right\|^2\right] \leq \frac{L^2}{\tau} \sum_{r=0}^{\tau-1} \mathbb{E}\left[\left\|\boldsymbol{\theta}^{(t,0)} - \boldsymbol{\theta}_k^{(t,r)}\right\|^2\right] \tag{42}$$

$$\leq (\tau-1)L^2\eta^2\sigma^2 + (\tau-1)L^2\eta^2 \underbrace{\sum_{s=0}^{\tau-1} \mathbb{E}\left[\left\|\nabla F_k(\boldsymbol{\theta}_k^{(t,s)})\right\|^2\right]}_{N_6}, \tag{43}$$

where $N_6$ in (43) can be further bounded.

### E.2.6  BOUNDING $N_6$ IN (43)

$$\mathbb{E}\left[\left\|\nabla F_k(\boldsymbol{\theta}_k^{(t,s)})\right\|^2\right]$$

$$\leq 2\mathbb{E}\left[\left\|\nabla F_k(\boldsymbol{\theta}_k^{(t,s)}) - \nabla F_k(\boldsymbol{\theta}^{(t,0)})\right\|^2\right] + 2\mathbb{E}\left[\left\|\nabla F_k(\boldsymbol{\theta}^{(t,0)})\right\|^2\right] \tag{44}$$

$$\leq 2L^2\mathbb{E}\left[\left\|\boldsymbol{\theta}^{(t,0)} - \boldsymbol{\theta}_k^{(t,s)}\right\|^2\right] + 2\mathbb{E}\left[\left\|\nabla F_k(\boldsymbol{\theta}^{(t,0)})\right\|^2\right], \tag{45}$$

where (44) uses $\|a + b\|^2 \leq 2\|a\|^2 + 2\|b\|^2$, (45) uses Lipschitz-smooth property. Plug (45) back to (43), we have

$$\frac{L^2}{\tau} \sum_{r=0}^{\tau-1} \mathbb{E}\left[\left\|\boldsymbol{\theta}^{(t,0)} - \boldsymbol{\theta}_k^{(t,r)}\right\|^2\right]$$

$$\leq (\tau-1)L^2\eta^2\sigma^2 + 2(\tau-1)\eta^2 L^4 \sum_{s=0}^{\tau-1} \mathbb{E}\left[\left\|\boldsymbol{\theta}_k^{(t,0)} - \boldsymbol{\theta}^{(t,s)}\right\|^2\right]$$

$$+ 2(\tau-1)\eta^2 L^2 \sum_{s=0}^{\tau-1} \mathbb{E}\left[\left\|\nabla F_k(\boldsymbol{\theta}^{(t,0)})\right\|^2\right] \tag{46}$$

After rearranging, we have the following bound for $N_4$:

$$\mathbb{E}\left[\left\|\nabla F_k(\boldsymbol{\theta}^{(t,0)}) - \mathbf{h}_k\right\|^2\right] \leq \frac{L^2}{\tau}\sum_{r=0}^{\tau-1}\mathbb{E}\left[\left\|\boldsymbol{\theta}^{(t,0)} - \boldsymbol{\theta}_k^{(t,r)}\right\|^2\right] \tag{47}$$

$$\leq \frac{(\tau-1)\eta^2\sigma^2L^2}{1-2\tau(\tau-1)\eta^2L^2} + \frac{2\tau(\tau-1)\eta^2L^2}{1-2\tau(\tau-1)\eta^2L^2}\mathbb{E}\left[\left\|\nabla F_k(\boldsymbol{\theta}^{(t,0)})\right\|^2\right] \tag{48}$$

$$= \frac{(\tau-1)\eta^2\sigma^2L^2}{1-C} + \frac{C}{1-C}\mathbb{E}\left[\left\|\nabla F_k(\boldsymbol{\theta}^{(t,0)})\right\|^2\right], \tag{49}$$

where we define $C = 2\tau(\tau-1)\eta^2L^2 < 1$. Then, the last term in equation 31 can be bounded as:

$$\tau\eta\sum_{k=1}^{K}w_k\mathbb{E}\left[\left\|\nabla F_k(\boldsymbol{\theta}^{(t,0)}) - \mathbf{h}_k\right\|^2\right]$$

$$\leq \tau\eta\sum_{k=1}^{K}\left\{w_k\left[\frac{(\tau-1)\eta^2\sigma^2L^2}{1-C} + \frac{C}{1-C}\mathbb{E}\left[\left\|\nabla F_k(\boldsymbol{\theta}^{(t,0)})\right\|^2\right]\right]\right\} \tag{50}$$

$$\leq \frac{\tau(\tau-1)\sigma^2L^2\eta^3}{1-C} + \frac{\tau\eta AC}{1-C}\mathbb{E}\left[\left\|\nabla F(\boldsymbol{\theta}^{(t,0)})\right\|^2\right] + \frac{\tau\eta BC}{1-C}, \tag{51}$$

where (51) follows the bounded dissimilarity assumption in Assumption 4. Plug (51) back to (31), we have

$$\mathbb{E}\left[F(\boldsymbol{\theta}^{(t+1,0)})\right] - F(\boldsymbol{\theta}^{(t,0)})$$

$$\leq -\frac{\tau\eta(1-AW_D)}{2}\left\|\nabla F(\boldsymbol{\theta}^{(t,0)})\right\|^2 + L\tau\eta^2\sigma^2\sum_{k=1}^{K}w_k^2 + \frac{\tau\eta BW_D}{2} + \tau\eta\sum_{k=1}^{K}w_k\mathbb{E}\left[\left\|\nabla F_k(\boldsymbol{\theta}^{(t,0)}) - \mathbf{h}_k\right\|^2\right]$$

$$\leq -\frac{\tau\eta(1-W_D)}{2}\left\|\nabla F(\boldsymbol{\theta}^{(t,0)})\right\|^2 + L\tau\eta^2\sigma^2\sum_{k=1}^{K}w_k^2 + \frac{\tau\eta BW_D}{2}$$

$$+ \frac{\tau(\tau-1)\sigma^2L^2\eta^3}{1-C} + \frac{\tau\eta AC}{1-C}\mathbb{E}\left[\left\|\nabla F(\boldsymbol{\theta}^{(t,0)})\right\|^2\right] + \frac{\tau\eta BC}{1-C} \tag{52}$$

$$= -\frac{\tau\eta}{2}\left(1 - W_D - \frac{2AC}{1-C}\right)\left\|\nabla F(\boldsymbol{\theta}^{(t,0)})\right\|^2 + L\tau\eta^2\sigma^2\sum_{k=1}^{K}w_k^2 + \frac{\tau\eta BW_D}{2} + \frac{\tau(\tau-1)\sigma^2L^2\eta^3}{1-C} + \frac{\tau\eta BC}{1-C} \tag{53}$$

Finally, by taking the average expectation across all rounds, we finish the proof of Theorem 1.

$$\min_t\mathbb{E}\left\|\nabla F(\boldsymbol{\theta}^{(t,0)})\right\|^2 \leq \frac{1}{T}\sum_{t=0}^{T-1}\mathbb{E}\left\|\nabla F(\boldsymbol{\theta}^{(t,0)})\right\|^2 \tag{54}$$

$$\leq \frac{2(1-C)\left(F(\boldsymbol{\theta}^{(0,0)}) - F_{inf}\right)}{\tau\eta T\left[1 - C - 2AC - W_D(1-C)\right]} + \frac{(1-C)BW_D}{\left[1 - C - 2AC - W_D(1-C)\right]}$$

$$+ \frac{2(1-C)L\eta\sigma^2\sum_{k=1}^{K}w_k^2}{\left[1 - C - 2AC - W_D(1-C)\right]} + \frac{2(\tau-1)\sigma^2L^2\eta^2}{\left[1 - C - 2AC - W_D(1-C)\right]} + \frac{2BC}{\left[1 - C - 2AC - W_D(1-C)\right]} \tag{55}$$

$$= \frac{1}{1 - C - 2AC - W_D(1-C)}\left(\underbrace{\frac{2(1-C)\left(F(\boldsymbol{\theta}^{(0,0)}) - F_{inf}\right)}{\tau\eta T}}_{T_1} + \underbrace{(1-C)BW_D}_{T_2}\right.$$

$$\left. + \underbrace{2(1-C)L\eta\sigma^2\sum_{k=1}^{K}w_k^2}_{T_3} + \underbrace{2(\tau-1)\sigma^2L^2\eta^2}_{T_4} + \underbrace{2BC}_{T_5}\right), \tag{56}$$

where $W_D = 2K \left[ \sum_{k=1}^{K} (p_k - w_k)^2 \right]$, $w_k$ is the aggregation weight, $p_k$ is the coefficient of global objective function, $C = 2\tau(\tau - 1)\eta^2 L^2 < 1$, $\tau$ is the number of steps in local model training, $\eta$ is learning rate, $T$ is the total communication round in FL, $K$ is the total client number, $F_{inf}, A, B, L, \sigma$ are the constants in assumptions.

### E.3 PROOF OF LEMMA 1

*Suppose $\{A_t\}_{t=1}^{T}$ is a sequence of random matrices and follows $\mathbb{E}[A_t | A_{t-1}, A_{t-2}, ..., A_1] = \mathbf{0}$, then*

$$\mathbb{E}\left[ \left\| \sum_{t=1}^{T} A_t \right\|_F^2 \right] = \sum_{t=1}^{T} \mathbb{E}\left[ \|A_t\|_F^2 \right]$$

*Proof.*

$$\mathbb{E}\left[ \left\| \sum_{t=1}^{T} A_t \right\|_F^2 \right] = \sum_{t=1}^{T} \mathbb{E}\left[ \|A_t\|_F^2 \right] + \sum_{i=1}^{T} \sum_{j=1, j \neq i}^{T} \mathbb{E}\left[ Tr\{A_i^\top A_j^\top\} \right] \tag{57}$$

$$= \sum_{t=1}^{T} \mathbb{E}\left[ \|A_t\|_F^2 \right] + \sum_{i=1}^{T} \sum_{j=1, j \neq i}^{T} Tr\{\mathbb{E}\left[ A_i^\top A_j^\top \right]\} \tag{58}$$

$$= \sum_{t=1}^{T} \mathbb{E}\left[ \|A_t\|_F^2 \right], \tag{59}$$

where (59) comes from assuming $i < j$ and using the law of total expectation $\mathbb{E}\left[ A_i^\top A_j \right] = \mathbb{E}\left[ A_i^\top \mathbb{E}[A_j | A_i, ..., A_1] \right] = \mathbf{0}$.

