# OpenReview forum: "Customizing Global Model for Diverse Target Distributions in Federated Learning"
_ICLR.cc/2024/Conference — ICLR 2024 Conference Withdrawn Submission_

### Official Review · Reviewer_oAHD · 2023-10-30

**Soundness:** 2 fair
**Presentation:** 3 good
**Contribution:** 2 fair
**Rating:** 3
**Confidence:** 3

**Summary:**

-This paper studies federated learning, with a specific focus on training a global model that can adapt to arbitrary target distribution. The authors propose a self-supervised learning based approach that aggregates the client models through a reweighting mechanism, where the weights are learned by maximizing a similarity metric (cosine similarity in the paper) between predictions generated from different transformation views (e.g., varying light conditions). They also conduct experiments on standard ML benchmark datasets and provide an ablation study.

**Strengths:**

-The paper is well-written and easy to follow. The authors conduct a thorough literature review and provide a comparison of their proposed method against a substantial set of baseline approaches.

**Weaknesses:**

-My first concern is about the novelty of the paper given that using self-supervised learning ideas to tackle the non-IID data challenges in federated learning is not new, for example, see the following:

Divergence-aware federated self-supervised learning, ICLR 2022

Federated Self-supervised Learning for Heterogeneous Clients. arXiv 2022

-The title of the paper (given by the paper pdf but not the one on the openreview page) places an emphasis on “arbitrary target distribution” which appears to be a bit over-claimed to me. In fact, the paper only empirically demonstrates that the proposed FedSSA method works for a few heterogeneous settings where the target distribution undergoes some standard distribution shifts, i.e., class-prior shift (see definitions in Dataset Shift in Machine Learning, The MIT Press).

There is no theoretical analysis on how the proposed FedSSA method can cater **arbitrary target distribution**, or even to what extent the proposed FedSSA method can be robust to the target distribution shift. In Appendix A the authors analyzed the convergence of the proposed method, however, with an extreme target distribution shift, Assumption A.4 can never hold. Think of the target distribution shift problem in standard centralized ML, generalization to the target distribution always depends on the difference of cross-domain distributions and cannot work for arbitrary target distributions (see the following paper for a reference). Therefore the analysis for FedAvg **without** target distribution shift cannot work for the analysis for FedSSA **with** distribution shift.

Generalizing to Unseen Domains: A Survey on Domain Generalization, TKDE 2022

**Questions:**

-I wonder the rationale behind employing a reweighting-based approach for aggregating client models. Is there a particular reason for not considering a sub-sampling strategy to select clients for aggregation in each round instead?

-How can the novelty of the current paper be justified in the context of previous research that has also explored the application of self-supervised learning ideas to address non-IID data challenges in federated learning?

-What are the theoretical guarantees that support the proposed FedSSA method? How can it be ensured that this method is effective for a wide range of target distributions, even **arbitrary target distributions**?

---

### Official Review · Reviewer_Q6HC · 2023-11-01

**Soundness:** 2 fair
**Presentation:** 2 fair
**Contribution:** 2 fair
**Rating:** 3
**Confidence:** 4

**Summary:**

This study introduces a novel Federated Learning (FL) setup referred to as "Pragmatic Federated Learning." In this setup, the primary objective of the server is to develop a global model that is specifically geared towards the unlabeled data stored on the server. This is accomplished through the "Self-Supervised Aggregation (FedSSA)" method.

Under FedSSA, the server maintains fixed local model parameters while performing two key tasks. First, it aggregates the predictions made by the individual local models. Second, it learns aggregation weights using a self-supervised loss function, which is applied to the unlabeled dataset residing on the server. This approach enables the server to iteratively refine and enhance the global model towards the unlabeled data.

**Strengths:**

* The writing is clear in the description
* The experiments provide enough details and ablation studies.

**Weaknesses:**

* I am not very convinced by the motivation of the new proposed setup. The setting assumes that the decentralized labeled data and the server unlabeled data are from the same domain but just differ in their label distribution, at least as suggested by the current experiments. This doesn't sound very well motivated for practical applications. Even in centralized learning literature, I did not find much research on this; stronger motivation will be needed for pragmatic  FL.

* The proposed approach is a bit disconnected from the setup. If only the label distributions are different, likely it does not need to change the whole model but only calibrate the classifier based on the target label distribution p(y). It is not obvious to me why should we learn the aggregation weights.

* The idea of leveraging unlabeled data in the server for distillation or aggregation is the most relevant technique, but many previous works are missing in comparison and discussion such as [i, ii, iii]

[i] Decentralized Learning with Multi-Headed Distillation, CVPR 2023
[ii] Data-Free Knowledge Distillation for Heterogeneous Federated Learning, ICML 2021
[iii] FedBE: Making Bayesian model ensemble applicable to federated learning, ICLR 2020

**Questions:**

* Since it is in a federated setting, how can we know if the server dataset and the decentralized data share enough similarity to ensure the pseudo labels are meaningful? What if they are from different domains?

* Since the target data are actually seen on the server, what is the difference in test time adaptation applied to FL?

---

### Official Review · Reviewer_ovj7 · 2023-11-04

**Soundness:** 3 good
**Presentation:** 3 good
**Contribution:** 2 fair
**Rating:** 3
**Confidence:** 5

**Summary:**

The author consider FL to address an unknown target distribution and propose a self-supervised aggregation scheme that satisfies transformation invariance. The authors provide substantial experiments on $12$ datasets.

I have substantial concerns regarding the developed method and lack of any sort of guarantees, which shows there is a mismatch between the claims in the paper and the actual results.

**Strengths:**

Providing experiments on several datasets and a number of classical FL baselines are among strengths of this paper.

**Weaknesses:**

One major issue with the considered setting in this paper is the underlying assumption that there is a unique and fixed target distribution ${\cal D}_T$. In fact, each client may have their own target distribution which may evolve over time.

----------

The formulation (2) is unclear. The objective is over $w_k$'s while ${\theta_k}$'s are constant for $k=\{1,\cdots,k\}$ . This objective does not provide any guarantees in terms of closeness of the solution to the solution of Eq (1).

The heuristic solution in Eq. (3) also does not provide any guarantees for the solution in Eq. (1).

----------

There is "Theorem 1" proof in Appendix E but I cannot find the theorem statement. The paper is not backed by theoretical results since the proposed method is just a heuristics without any sort of provable connection between the developed solution and the original objective minimizer in Eq. (1).

----------

Overall, I think this paper lacks of any sort of guarantees, which shows there is a mismatch between the claims in the paper that FedSSA addresses adaptation to diverse target datasets and the actual results.

**Questions:**

Please see above.

---

### Official Review · Reviewer_6Ffb · 2023-11-05

**Soundness:** 3 good
**Presentation:** 3 good
**Contribution:** 3 good
**Rating:** 6
**Confidence:** 4

**Summary:**

The paper proposes FedSSA, a novel federated learning approach that addresses the scenario where the target distribution may be different from the true one and the target data is unlabeled. FedSSA uses a self-supervised aggregation method that adjusts to specific, arbitrary data distributions of the target dataset, improving model performance. It dynamically adjusts aggregation weights for local models to ensure transformation invariance. The method is tested on 4 benchmark datasets.

**Strengths:**

1. The paper studies a practical and novel problem, where the target distribution may be different from the training distribution, and the target data distribution is unlabeled.
2. The proposed three terms are well-motivated, and the performance is verified by the experiments on popular benchmark datasets.

**Weaknesses:**

The main weakness is the lack of theoretical guarantees. The three essential spirits in Section 4.2 help readers understand why the proposed method may work. However, theoretical analysis is necessary to prove when the proposed method works, and when the proposed method may fail. It is important when the target label distribution is unlabeled.

**Questions:**

In Table 1, when target distributions are imbalanced, how is the accuracy calculated? Should it be an average weighted by the sample frequency of each label class?